# Regulatory changes in two chemoreceptor genes contribute to a *Caenorhabditis elegans* QTL for foraging behavior

Joshua S Greene[1], May Dobosiewicz[1], Rebecca A Butcher[2], Patrick T McGrath[3], Cornelia I Bargmann[1]*

[1]Howard Hughes Medical Institute (HHMI), Lulu and Anthony Wang Laboratory of Neural Circuits and Behavior, The Rockefeller University, New York, United States; [2]Department of Chemistry, University of Florida, Gainesville, United States; [3]Department of Biology, Georgia Institute of Technology, Atlanta, United States

**Abstract** Natural isolates of *C. elegans* differ in their sensitivity to pheromones that inhibit exploratory behavior. Previous studies identified a QTL for pheromone sensitivity that includes alternative alleles of *srx-43*, a chemoreceptor that inhibits exploration through its activity in ASI sensory neurons. Here we show that the QTL is multigenic and includes alternative alleles of *srx-44*, a second chemoreceptor gene that modifies pheromone sensitivity. *srx-44* either promotes or inhibits exploration depending on its expression in the ASJ or ADL sensory neurons, respectively. Naturally occurring pheromone insensitivity results in part from previously described changes in *srx-43* expression levels, and in part from increased *srx-44* expression in ASJ, which antagonizes ASI and ADL. Antagonism between the sensory neurons results in cellular epistasis that is reflected in their transcription of insulin genes that regulate exploration. These results and genome-wide evidence suggest that chemoreceptor genes may be preferred sites of adaptive variation in *C. elegans*.

*For correspondence: cori@rockefeller.edu

**Competing interests:** The authors declare that no competing interests exist.

## Introduction

Social communication, particularly chemical communication with pheromones, is broadly used throughout nature to organize behavior. Pheromones and the responses that they drive are highly diverse within and between species. The detection of diverse pheromones by large families of finely-tuned G protein-coupled receptors is well-recognized, but it is less clear how these different receptors drive unique, overlapping, and evolving behavioral and physiological responses (*Yang and Shah, 2016*).

The nematode *C. elegans* communicates through the secretion and detection of pheromones called ascarosides, which reflect population density and vary with an animal's sex, developmental stage, and feeding status (*Izrayelit et al., 2012*; *Jeong et al., 2005*; *Kaplan et al., 2011*). At least four classes of sensory neurons detect ascarosides; these can induce entry into and exit from the developmental dauer stage, aggregation into feeding groups, male attraction to hermaphrodite mating partners, or acute avoidance (*Jang et al., 2012*; *Kim et al., 2009*; *McGrath et al., 2011*; *Srinivasan et al., 2008*). In addition, a subset of ascarosides potently modifies innate foraging behaviors. On a lawn of bacterial food, *C. elegans* spontaneously alternates between minute-long foraging states called roaming and dwelling (*Ben Arous et al., 2009*; *Flavell et al., 2013*; *Fujiwara et al., 2002*). Roaming animals move quickly to explore a large area, while dwelling animals

move slowly and reverse frequently to exploit local food resources. A number of ascarosides shorten roaming states to suppress exploration at high population density, including the indolated ascaroside icas#9 (also known as IC-asc-C5 or C5)(*Butcher et al., 2009*; *Greene et al., 2016*). icas#9 is detected by the G protein-coupled chemoreceptor SRX-43 in ASI sensory neurons, which are components of the distributed neuromodulatory circuit that regulates roaming and dwelling (*Flavell et al., 2013*; *Greene et al., 2016*).

Some wild-type strains including the German strain MY14 have reduced sensitivity to icas#9 due to variation at *roam-1,* a 37 kb Quantitative Trait Locus (QTL) that includes the *srx-43* gene (*Greene et al., 2016*). icas#9 insensitivity results in part from reduced *srx-43* expression in MY14 compared to N2. Both MY14-like and N2-like *roam-1* alleles are distributed across four continents, and population genetic studies suggest that the *roam-1* QTL is under balancing selection, such that both alleles are actively maintained in wild populations. Although the natural ecology of *C. elegans* is incompletely understood, laboratory competition studies indicate that alternative *roam-1* alleles can be favored depending on the presence of pheromones and the distribution of food, consistent with balancing selection on this locus.

The genetic architecture of individual differences in behavior is a central element of neurogenetics that is only beginning to be understood (*Greenspan, 2009*). Many studies have suggested that individual mutations with small effect sizes interact to generate natural variation, but counterexamples of large-effect single genes have been described as well (*Buchner and Nadeau, 2015*; *Greenspan, 2009*). The contribution of *srx-43* to foraging appeared to represent a case in which a single gene has a large effect on a behavioral trait, as the *roam-1* QTL explains about 40% of the genetic variance in icas#9 sensitivity. Here we show that the previously identified *roam-1* QTL is more complex than suggested by initial studies: it reflects changes both in *srx-43* and in an adjacent gene, *srx-44*. The reduced icas#9 sensitivity in strains bearing the *roam-1*$_{MY14}$ QTL arises from reduced *srx-43* expression in ASI neurons, and acquisition of *srx-44* expression in ASJ sensory neurons. Natural behavioral variation results from the remapping of these two chemoreceptor genes across multiple sensory neurons with antagonistic actions.

## Results

### *srx-44* increases icas#9 sensitivity in N2 and decreases sensitivity in *roam-1*$_{MY14}$

The ascaroside pheromone icas#9 strongly suppresses exploration behavior in the N2 laboratory strain and in most wild strains, as determined by measuring the area a single animal explores on a bacterial lawn in an overnight assay (*Figure 1*). By contrast, icas#9 does not suppress exploration in animals from the wild MY14 strain or in the Near Isogenic Line (NIL) *roam-1*$_{MY14}$, which bears a 182 kb region from MY14 in an otherwise N2 genetic background (*Greene et al., 2016*) (*Figure 2A,B*). The *srx-43* gene, which is strongly expressed in N2 and weakly expressed in MY14, is an essential element of the *roam-1* QTL (*Greene et al., 2016*). We found that the N2 *roam-1* allele was recessive to the MY14 allele: F1 progeny of N2 animals crossed with the *roam-1*$_{MY14}$ NIL were insensitive to icas#9 (*Figure 2B*). However, the F1 progeny of N2 animals crossed with an *srx-43* null mutant were sensitive to icas#9, like N2, indicating that an *srx-43(lf)* mutation is recessive (*Figure 2B*). The dominance of *roam-1*$_{MY14}$ over N2 indicates that the *roam-1* QTL is not entirely explained by reduced *srx-43* expression in *roam-1*$_{MY14}$.

These results prompted more detailed analysis of the *roam-1* QTL. The endogenous *srx-43* locus was inactivated by null mutations in N2 and in the *roam-1*$_{MY14}$ N2 NIL, resulting in profound icas#9 insensitivity, and rescued with a single-copy N2 *srx-43* gene targeted to a different chromosome. The N2 *srx-43* single copy transgene restored icas#9 sensitivity to both N2 and *roam-1*$_{MY14}$ *srx-43* mutants (*Figure 2C*), but the *roam-1*$_{MY14}$ strain was less sensitive to icas#9 than its N2 counterpart (*Figure 2C*). As both of these strains bear the same N2 *srx-43* transgene, these experiments confirm that the difference in *srx-43* function cannot fully account for the *roam-1* behavioral phenotype.

The *srx-44* gene was an attractive candidate for a second locus, as *srx-43* and *srx-44* are adjacent, closely-related paralogs in a large chemoreceptor gene family that might be expected to have related functions. In addition, both genes fall within a 20 kb region of the minimal 37 kb *roam-1* QTL that has exceptionally high sequence divergence between N2 and MY14. Protein-terminating *srx-44*

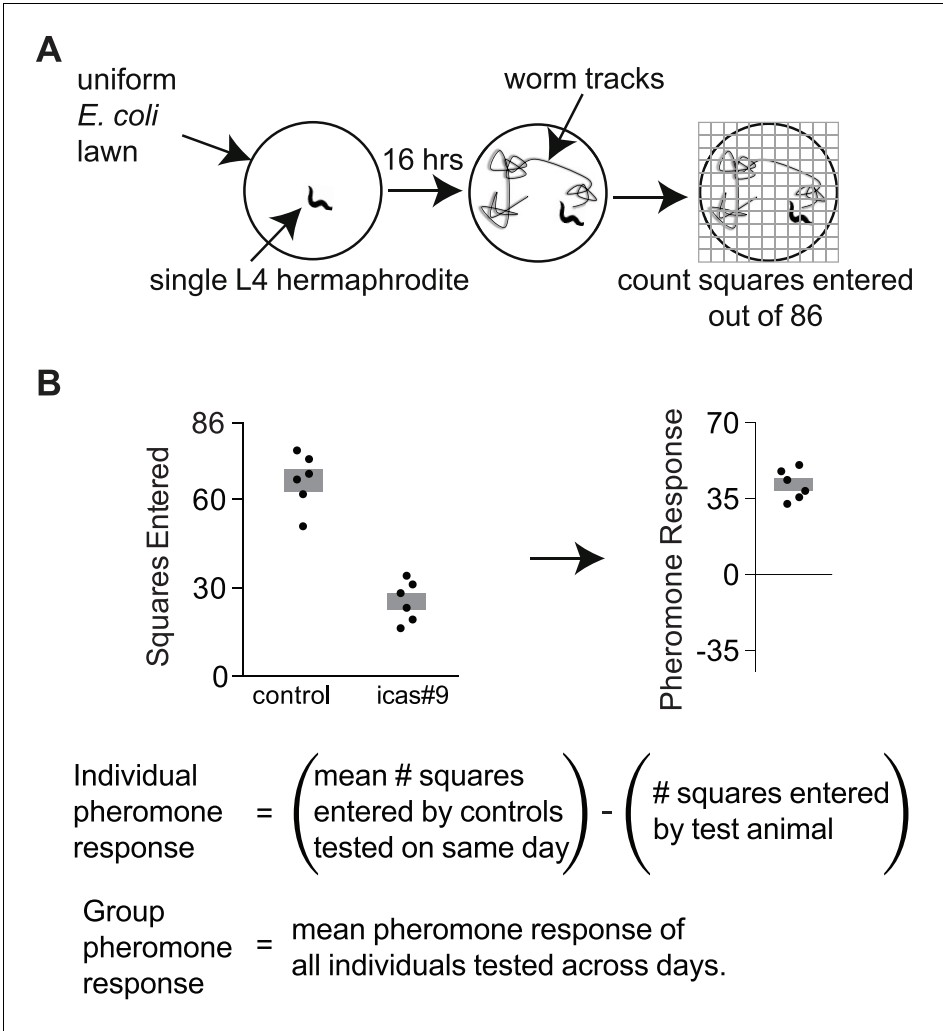

**Figure 1.** Exploration assays. (**A**) Individual animals are allowed to explore a thinly seeded 3.5 cm plate for 16 hr, after which exploration is scored by placing the plate on a grid and counting squares with tracks (*Flavell et al., 2013*). Diagram from *Greene et al., 2016*. (**B**) A pheromone response for each animal on an ascaroside plate was determined with respect to the behavior of control animals that were tested on ascaroside-free plates on the same day. Data are representative values for one day of testing; boxes indicate the mean ± SEM. Below, calculating individual and group pheromone response.

The following source data is available for figure 1:

**Source data 1.** Individual exploration assays in *Figure 1B*.

*(lf)* alleles were generated by CRISPR-cas9 mutagenesis in N2 and *roam-1*$_{MY14}$ strains. In each case, *srx-44(lf)* resulted in an icas#9 response that was intermediate between those of N2 and *roam-1*$_{MY14}$ (*Figure 2D*). Since N2 became less icas#9 sensitive in the *srx-44* null, and *roam-1*$_{MY14}$ became more sensitive, the alternative *srx-44* alleles have opposite effects on icas#9 sensitivity.

The effects of *srx-44(lf)* mutations were weaker than those of *srx-43(lf)* (*Figure 2C,D*), which suggest that *srx-44* acts a modifier gene. We examined this possibility by generating a *roam-1*$_{MY14}$ *srx-43(lf) srx-44(lf)* strain by CRISPR-Cas9 mutagenesis. The resulting double mutant was insensitive to icas#9, like *srx-43(lf)* (*Figure 2D*). These results suggest that *srx-43* is essential for icas#9 sensitivity in all genetic backgrounds, whereas *srx-44* modifies icas#9 response in opposite directions depending on the *roam-1* allele.

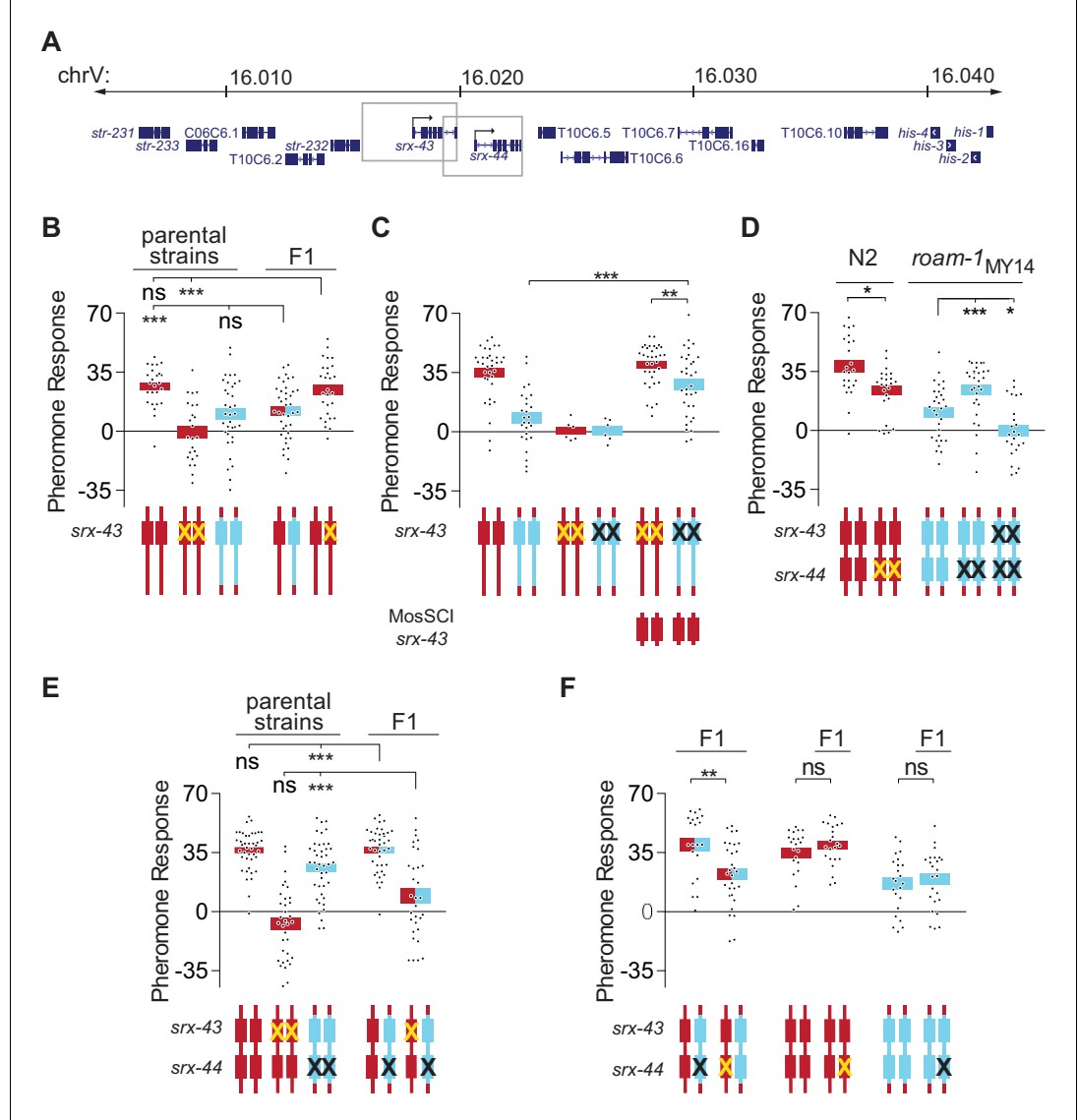

**Figure 2.** *srx-43* and *srx-44* both influence icas#9 sensitivity.  **(A)** The *roam-1* locus. Boxes indicate genomic regions used for *srx-43* and *srx-44* transgenes. **(B)** Dominance tests. Pheromone response of parental strains and of the F1 progeny from crosses between N2 and *roam-1*_MY14 or between N2 and N2 *srx-43(lf)*. **(C)** *srx-43* variation is insufficient to explain the *roam-1* QTL. **(D)** Pheromone response of *srx-44* loss-of-function mutants. **(E)** Complementation test between *srx-43* and *roam-1*_MY14. Pheromone response of parental strains and of F1 progeny from crosses between *roam-1*_MY14 *srx-44(lf)* and N2 or N2 *srx-43(lf)*. **(F)** Reciprocal hemizygosity test for *srx-44*. Left, pheromone response of the F1 progeny from crosses between N2 and *roam-1*_MY14 *srx-44(lf)* and between N2 *srx-44(lf)* and *roam-1*. Center and right, hemizygosity for *srx-44* in the parental strains did not affect behavior. For Figure B–F, boxes indicate the mean pheromone response ± SEM. Box color indicates genotype at the *roam-1* locus (Red = N2; Blue = MY14; red and blue = heterozygous). Cartoons of the *roam-1* locus show endogenous *srx-43* (B–F) and *srx-44* (D–F) with the same color code. X indicates null allele. In C, 'MosSCI *srx-43*' indicates strains with a chromosome II Mos1 Single Copy Inserted *srx-43* allele from N2.. ***p<0.001,**p<0.01, *p<0.05; ns, not significant by ANOVA with Dunnett correction (A, D- *roam-1*_MY14, E) or t-test (C, D- N2, F).

The following source data is available for figure 2:

**Source data 1.** Individual exploration assays in *Figure 2B–F*.

With this additional information, genetic tests were conducted to assess the contributions of *srx-43* and *srx-44* to the *roam-1* QTL. The F1 progeny of a *roam-1*_MY14 *srx-44(lf)* cross with N2 resembled N2 (*Figure 2E*), indicating that *srx-44* is necessary for the dominance of the *roam-1*_MY14 QTL. The F1 progeny of a cross between *roam-1*_MY14 *srx-44(lf)* and N2 *srx-43(lf)* were insensitive to icas#9

(*Figure 2E*). The failure of the *roam-1*$_{MY14}$ *srx-43* allele to complement N2 *srx-43(lf)* confirmed reduced *srx-43* function in *roam-1*$_{MY14}$. To test for altered *srx-44* function, a reciprocal hemizygosity test was conducted: The F1 progeny of N2 *srx-44(lf)* crossed with *roam-1*$_{MY14}$ were compared to the F1 progeny of N2 crossed with *roam-1*$_{MY14}$ *srx-44(lf)*. The resulting hemizygotes differed genetically only in whether their single functional *srx-44* allele derived from N2 or *roam-1*$_{MY14}$ and demonstrated significantly different icas#9 sensitivity (*Figure 2F*), indicating that *srx-44* variation in the *roam-1* QTL affects foraging behavior.

## Sequence variation in the *srx-44* promoter alters pheromone response

As existing sequence resources underestimated the exceptionally high sequence divergence between N2 and MY14 *srx-43* alleles (19.7%), we examined *srx-44* by Sanger resequencing. This analysis indicated that MY14 and N2 *srx-44* sequences differed by 5.0% across the 2.8 Kb promoter and coding region of the *srx-44* gene, approximately 25 times the genome-wide average (*Thompson et al., 2015*). A phylogeny of *srx-44* and related genes confirmed that *srx-44* in N2 and MY14 are alleles of one orthologous gene (*Figure 3F*). Despite this high level of divergence, only five polymorphisms alter amino acids between N2 and MY14 *srx-44* alleles, suggesting that the coding region remains under purifying selection (dN/dS = 0.071; five non-synonymous mutations, 20 synonymous mutations).

To localize the biologically relevant sequence changes between N2 and MY14 *srx-44* genes, we tested transgenes with N2 and *roam-1*$_{MY14}$ sequences for their biological activity. Although *roam-1*$_{MY14}$ was dominant to N2 in an F1 cross, high-copy transgenes that expressed the N2 *srx-44* gene were able to confer pheromone sensitivity on the *roam-1*$_{MY14}$ strain (*Figure 3A and B*). This result suggests that the N2 and MY14 *srx-44* alleles have antagonistic activities. Transgenes containing an N2 *srx-44* gene with an early nonsense mutation or a MY14 *srx-44* gene did not confer pheromone sensitivity (*Figure 3B*). Exchanging the promoters and coding regions of *srx-44* showed that transgenes with the N2 *srx-44* promoter conferred icas#9 sensitivity to *roam-1*$_{MY14}$ regardless of whether the coding region was from N2 or MY14, whereas transgenes with the MY14 promoter did not enhance pheromone sensitivity (*Figure 3C*). Therefore, promoter sequences account for differential activity of N2 and MY14 *srx-44* genes in this assay.

N2 and MY14 differ at 35 of the 515 bases between the start codon of *srx-44* and the end of the adjacent *srx-43* gene. Nine of these changes cluster in a region 34–72 bp upstream of the start codon of *srx-44*. Exchanging just this proximal promoter sequence in N2 and MY14 *srx-44* transgenes switched their activity (*Figure 3D*). To confirm the biological importance of this potential regulatory site, we precisely exchanged the sequences 34–72 bp upstream of the *srx-44* start codon at the endogenous genomic loci of N2 and *roam-1*$_{MY14}$ using oligonucleotide-templated homologous recombination with CRISPR/Cas9. Introducing the MY14 *srx-44* proximal promoter element into the N2 genome reduced response to icas#9 (*Figure 3E*). Conversely, introducing the N2 proximal promoter element for *srx-44* strongly enhanced the icas#9 sensitivity of *roam-1*$_{MY14}$ (*Figure 3E*). These result match the genetic predictions for an exchange of N2 and *roam-1*$_{MY14}$ *srx-44* alleles, and localize altered icas#9 sensitivity to the proximal promoter sequences of *srx-44*.

## *srx-44* acts in different neurons to promote or inhibit icas#9 sensitivity

To understand how altered *srx-44* activity was conferred by the promoter sequence, we examined the expression of *srx-44* genomic sequences linked to the green fluorescent protein (GFP) in bicistronic transcripts. Transgenes bearing the N2 promoter sequence drove GFP expression selectively in the two ADL sensory neurons, while transgenes bearing the MY14 promoter sequence drove GFP in the ADL and ASJ sensory neurons (*Figure 4A*). The cell type specificity of *srx-44* expression was determined by the proximal promoter element that functionally differentiated N2 and MY14 *srx-44* genes: exchanging this promoter element between N2 and MY14 transgenes resulted in GFP reporter gene expression that matched the proximal element (ADL for N2, ADL and ASJ for MY14) (*Figure 4A*). These experiments localize both biological activity and expression to a proximal region 34–72 bp upstream of *srx-44*.

Since N2 and MY14 *srx-44* alleles have opposite effects on foraging behavior, both sites of expression are likely to be functionally important: *srx-44* might decrease icas#9 sensitivity when expressed in ASJ and increase icas#9 sensitivity when expressed in ADL. We tested this hypothesis

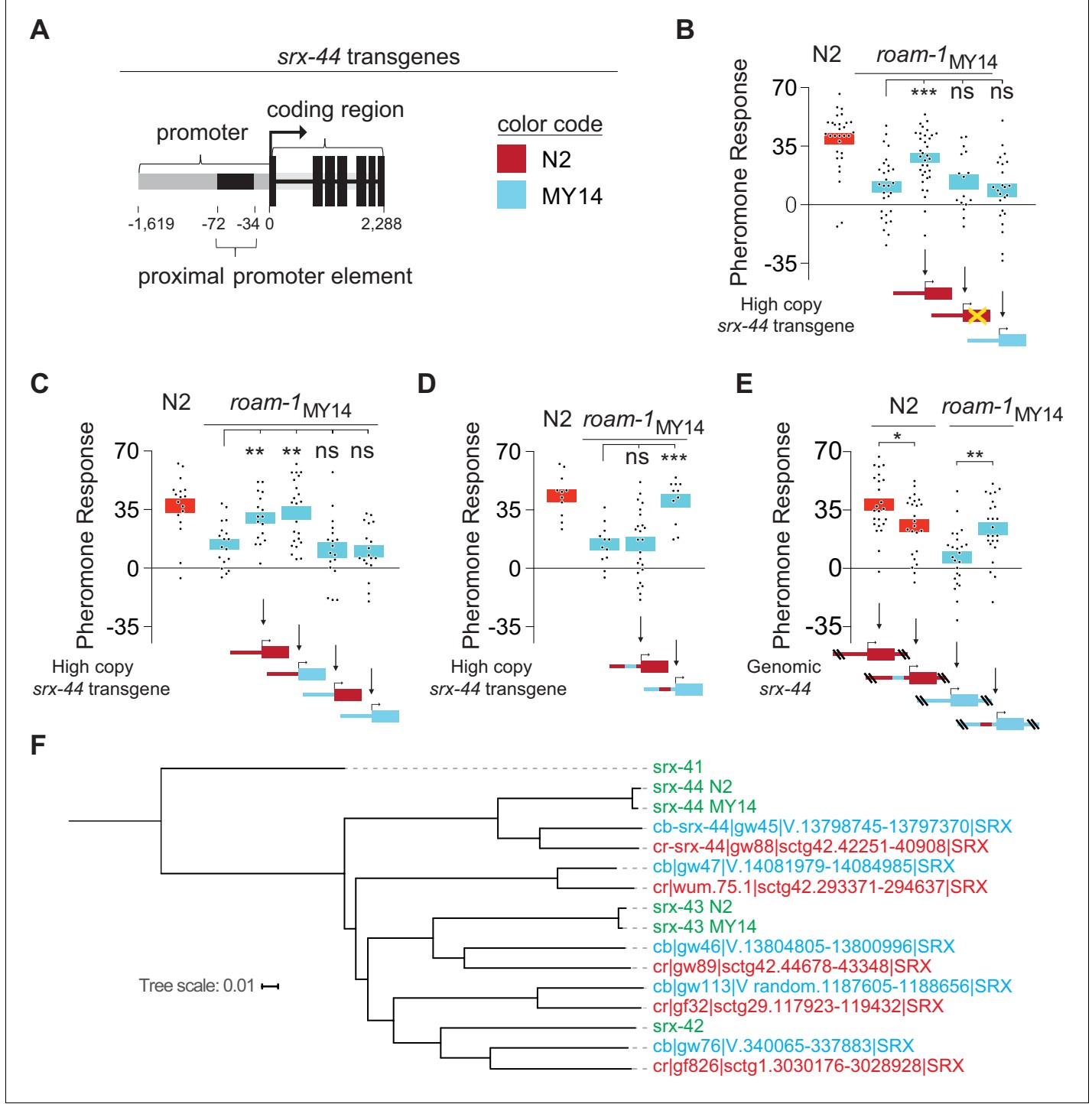

**Figure 3.** The proximal promoter sequence underlies altered *srx-44* activity in MY14. (A) Cartoon of *srx-44* transgenes (corresponds to grey box in *Figure 2A*). (B) N2 *srx-44* transgenes confer icas#9 sensitivity on *roam-1*<sub>MY14</sub>. Similar transgenes bearing a frameshift in the coding region do not, nor do MY14 *srx-44* transgenes. (C) *srx-44* transgenes with a N2 promoter confer icas#9 sensitivity on *roam-1*<sub>MY14</sub>, whereas *srx-44* transgenes with a MY14 promoter do not. (D) The proximal promoter element accounts for N2 *srx-44* activity in the transgene assay. (E) icas#9 responses of Allele Replacement Lines for the *srx-44* proximal promoter element, generated by homologous recombination in the endogenous genomic locus with CRISPR/Cas9, localize activity to the proximal promotor element. (F) Phylogeny constructed for the coding sequence of *srx-43*, *srx-44* and related genes in *C. elegans*, *C. briggsae*, and *C. remanei*. The *srx-44* alleles in N2 and MY14 are closely related, confirming they are alleles of a single gene. Genes are color coded by species (green = *C. elegans*, blue = *C. briggsae*, orange = *C. remanei*). Protein sequences and gene names are from ***Thomas and Robertson (2008)***. For B-E, boxes indicate the mean pheromone response ± SEM. Box color in the data panels indicates genotype at the genomic *roam-1* locus

*Figure 3 continued on next page*

*Figure 3 continued*

(Red = N2; Blue = MY14). Transgenes and allele replacements are indicated in cartoons with the same color code. ***p<0.001,**p<0.01,* p<0.05, ns = not significant by ANOVA with Dunnett correction (B, C, D) or by t test (E).

The following source data is available for figure 3:

**Source data 1.** Individual exploration assays in *Figure 3B–E*.

by expressing *srx-44* selectively in either ASJ or ADL neurons under the control of cell-specific promoters. Expression of *srx-44* under an ASJ-specific *srh-11* promoter reduced icas#9 responses in N2 (*Figure 4B*), supporting a suppressive activity in ASJ. Overexpression of *srx-44* under ADL-specific *sre-1* or *srh-220* promoters increased icas#9 responses in *roam-1*$_{MY14}$ (*Figure 4B*), supporting a

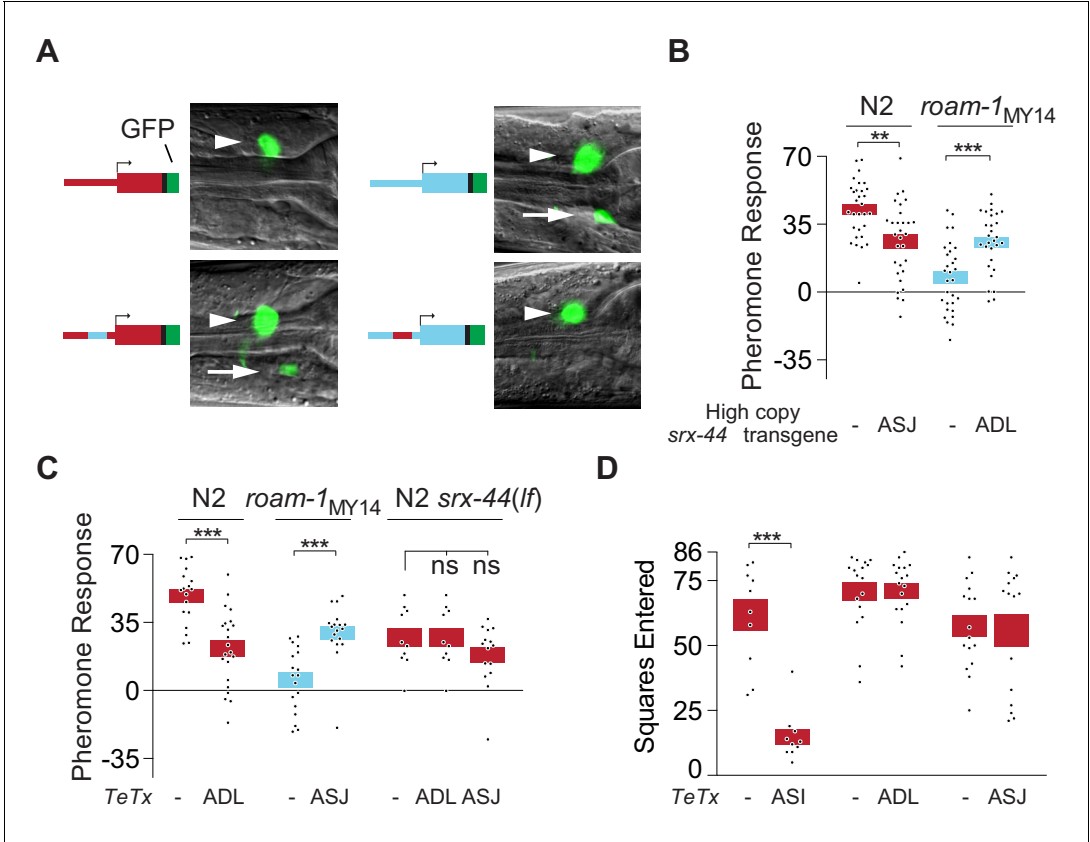

**Figure 4.** *srx-44* site of expression determines whether it potentiates (ADL) or suppresses (ASJ) behavioral response to icas#9. (**A**) Expression of GFP from *srx-44* reporter genes. Images show overlaid fluorescence and differential interference contrast images of the anterior ganglion, with anterior at the left and dorsal up. Arrowheads indicate ADL, arrows indicate ASJ. Cartoons show transgenes tested in each image. Top left = P*srx-44(N2)::GFP*; Top right = P*srx-44(MY14)::GFP*. Bottom left = P*srx-44(N2 distal promoter/MY14proximal promoter)::GFP*; Bottom right = P*srx-44(MY14 distal promoter/N2 proximal promoter)::GFP*. (**B**) Transgenes expressing *srx-44* under ASJ- or ADL-specific promoters oppositely affect pheromone responses. (**C**) Tetanus toxin light chain (TeTx) inhibition of neurotransmitter and neuropeptide release from ASJ or ADL affects icas#9 sensitivity in an *srx-44* dependent manner. (**D**) TeTx inhibition of ASI, but not ADL or ASJ, reduces exploration on pheromone-free control plates, expressed as mean squares entered ± SEM. *Boxes* indicate the mean ± SEM. *Box color* indicates genotype at the *roam-1* locus (Red = N2; Blue = MY14). All data presented as mean ± SEM. ***p<0.001,**p<0.01, ns = not significant by t test (B, C- N2 and *roam-1*$_{MY14}$, D) or by ANOVA with Dunnett correction (C-N2 *srx-44(lf)*).

The following source data is available for figure 4:

**Source data 1.** Individual exploration assays in *Figure 4B–D*.

positive role in ADL even in the presence of antagonistic ASJ expression in $roam-1_{MY14}$. This effect of ADL overexpression resembles the effect of high-copy N2 *srx-44* transgenes, which enhance icas#9 sensitivity in $roam-1_{MY14}$ (*Figure 3B*).

Neither the ADL neurons nor the ASJ neurons have previously been implicated in foraging behavior. As an independent way to interrogate their functions, we expressed the tetanus toxin light chain, which cleaves synaptobrevin to reduce neurotransmitter and neuropeptide secretion. ADL::tetanus toxin reduced the response to icas#9 in N2 (*Figure 4C*), indicating that vesicle release from ADL enhances N2 pheromone sensitivity. ASJ::tetanus toxin increased icas#9 responses in $roam-1_{MY14}$ (*Figure 4C*), indicating that vesicle release from ASJ antagonizes pheromone sensitivity. In *srx-44(lf)* mutants, neither ADL::tetanus toxin nor ASJ::tetanus toxin affected behavioral responses to icas#9, indicating that icas#9 acts through the SRX-44 chemoreceptor in ADL and ASJ. Notably, ASI::tetanus toxin reduced exploration in the absence of pheromones, whereas ADL::tetanus toxin or ASJ::tetanus toxin had no significant effect (*Figure 4D*). These results suggest that ASI is necessary for maintaining basal levels of exploration, in agreement with previous results (*Flavell et al., 2013*), whereas ADL and ASJ activity are relevant only in the presence of icas#9. These results also show that while *srx-43* activity in ASI promotes pheromone response and *srx-44* activity in ASJ inhibits pheromone response, both ASI and ASJ induce roaming through vesicle release (*Figure 5H*).

## Icas#9 influences behavior through TGF-β and insulin signaling pathways

Early in development, the ASI and ASJ neurons regulate the decision to enter the dauer larva stage by secreting insulin-related peptides whose transcription is regulated by ascarosides (*Li et al., 2003*). Insulin signaling also promotes roaming (*Ben Arous et al., 2009*), suggesting that insulins could be effectors of ASI and ASJ functions. To ask whether insulins are regulated by icas#9, we examined the expression of the insulin gene *daf-28* with integrated *daf-28::GFP* reporters that are expressed in both ASI and ASJ neurons. Animals were treated with icas#9 in the same protocol used in behavioral assays, and then examined using quantitative microscopy. In N2 animals, icas#9 decreased *daf-28::GFP* by 37% in ASI and by 28% in ASJ (*Figure 5A*). In $roam-1_{MY14}$ animals, icas#9 did not suppress *daf-28::GFP* expression, which also appeared elevated at baseline in the absence of pheromones (*Figure 5B*). Thus the expression of *daf-28::GFP* can be regulated by icas#9, with differential sensitivity in the N2 and MY14 alleles of the *roam-1* QTL.

With this molecular readout of icas#9 signaling, we examined the effects of the *srx-43* and *srx-44* genes on *daf-28* regulation. In N2 *srx-43(lf)* animals, icas#9 did not decrease *daf-28::GFP* in either ASI or ASJ, unlike N2 (*Figure 5C*). In $roam-1_{MY14}$ *srx-44(lf)* animals, icas#9 reduced *daf-28::GFP* expression in ASI neurons, although it did not have this effect in $roam-1_{MY14}$ (*Figure 5D*). Together these experiments show that *srx-43* and *srx-44* influence icas#9 regulation of *daf-28::GFP*, and follow the pattern that the N2 *roam-1* allele depends upon *srx-43*, and the MY14 *roam-1* allele depends upon *srx-44*. However, they are inconsistent with simple cell-autonomous action of each receptor: Comparing *Figure 5A and C* shows that *srx-43*, which is expressed only in ASI, affects *daf-28::GFP* expression in both ASI and ASJ. Comparing *Figure 5B* to *5D* shows that *srx-44* affects *daf-28::GFP* expression in ASI, where *srx-44* is not expressed. These results show that signaling between ASI and ASJ neurons affects their gene expression patterns (*Figure 5H*).

The influence of insulins on foraging behavior was further examined in insulin signaling mutants. Animals lacking *daf-2*, the receptor for all insulins, explored significantly less than N2 controls at baseline, precluding an analysis of their pheromone sensitivity (*Figure 5E*). Mutants bearing either the dominant interfering *daf-28(sa191)* insulin mutation or the recessive *daf-28(tm2308)* null allele resembled N2 in their exploration of control plates (*Figure 5E*), but responded only weakly to ascarosides compared to N2 (*Figure 5F*). Animals null for *daf-16*, which encodes a transcription factor that is inhibited by the insulin pathway, also responded weakly to icas#9 compared to N2 (*Figure 5F*). These results suggest that unidentified insulins and the DAF-2 insulin receptor regulate basal exploration, while *daf-28* acts in the ascaroside regulation of exploration.

In pheromone-regulated dauer larva development, insulin signaling converges with a parallel TGF-β signaling pathway to regulate a final common target, the nuclear hormone receptor DAF-12. Previous studies demonstrated that *srx-43* regulates the TGF-beta protein encoded by *daf-7*, and that the behavioral effects of icas#9 are partly suppressed in TGF-β signaling mutants, as they are in insulin signaling mutants (*Greene et al., 2016*). *daf-12(lf)* animals were profoundly insensitive to

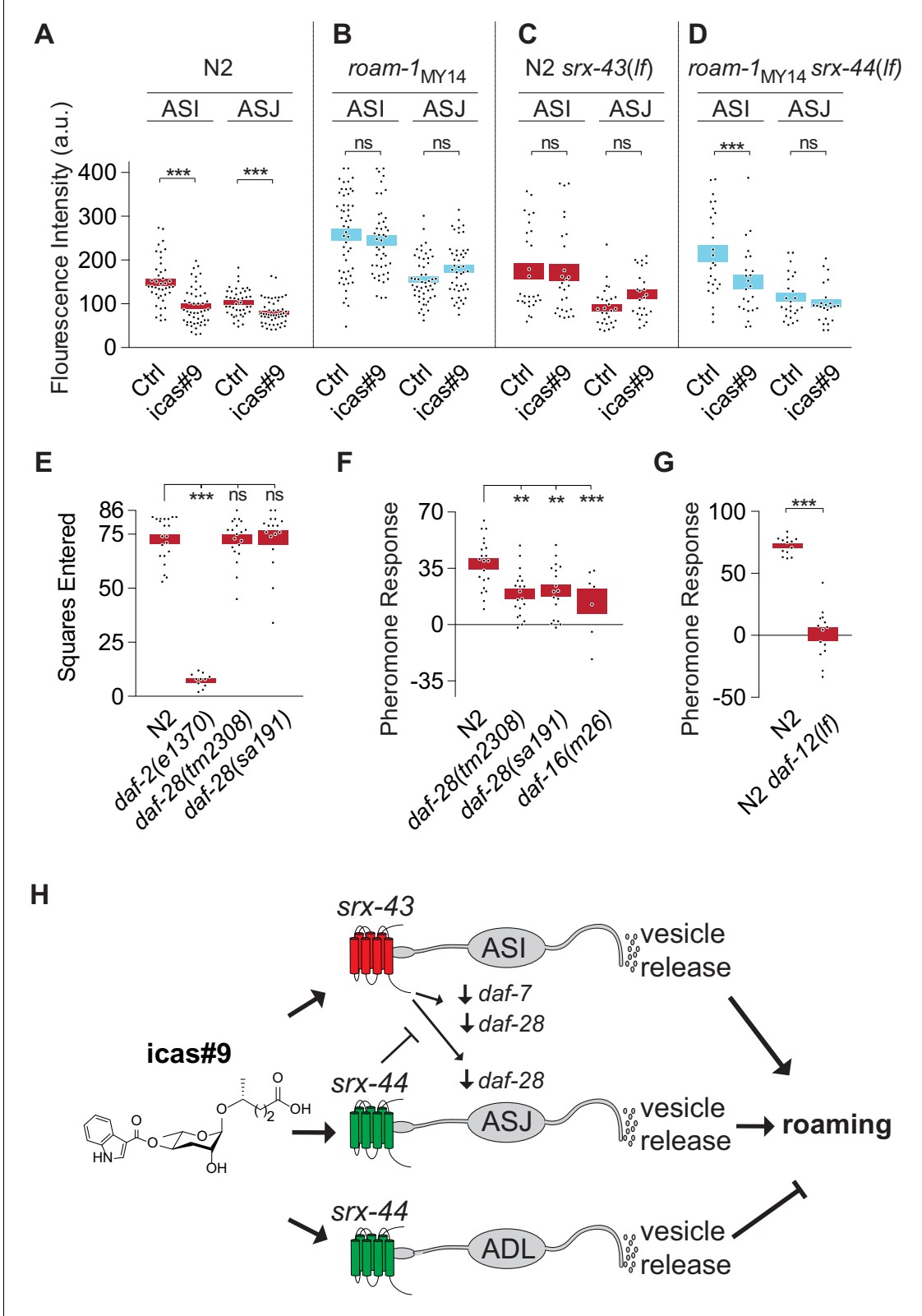

**Figure 5.** *srx-43* and *srx-44* regulate insulin and TGF-β endocrine signaling pathways. (A–D) Effect of icas#9 on ASI and ASJ *daf-28::GFP* expression in N2, *roam-1_MY14*, N2 *srx-43(lf)*, and *roam-1_MY14 srx-44(lf)* animals. Bars indicate mean fluorescence intensity ± SEM. (E) Exploration in the absence of pheromones in insulin pathway mutants, expressed as mean squares entered ± SEM. ***p<0.001, ns = not significant by ANOVA with Dunnett correction. (F) Pheromone response of insulin pathway mutants expressed as mean ± SEM. (G) *daf-12*, a convergence point of *daf-7* (TGF-beta) and *daf-*

*Figure 5 continued*

*28* (insulin) signaling in development, is necessary for icas#9 modulation of foraging behavior. Bars indicate icas#9 response expressed as mean ± SEM. (**H**) Schematic of the proposed relationships between icas#9, the chemoreceptors *srx-43* or *srx-44*, the sensory neurons ASI, ASJ or ADL, alterations in insulin and TGF-beta gene expression, vesicle release, and roaming behavior. SRX-43 and SRX-44 confer sensitivity to icas#9, which leads to changes in gene expression both within the sensing cell and within other sensory neurons. Activation of SRX-43 in ASI reduces the expression of *daf-7* (TGF-beta) and *daf-28* (insulin) genes as shown in *Figure 5A and C* and in *Greene et al. (2016)*. The activity of SRX-44 can antagonize this effect as shown in *Figure 5B and D*. Vesicle release from the sensory neurons, potentially releasing DAF-7, DAF-28, or other neurotransmitters or neuropeptides, can stimulate roaming at baseline (ASI; shown in *Figure 4D*), stimulate roaming in the presence of icas#9 (ASJ; *Figure 4C*), or inhibit roaming in the presence of icas#9 (ADL; *Figure 4C*). Boxes indicate the mean ± SEM, color indicates genotype at *roam-1* locus (red = N2, blue = MY14). ***p<0.001, **p<0.01, ns = not significant by t test (**A, D**) or by ANOVA with Dunnett correction (**B, C**).

The following source data and figure supplements are available for figure 5:

**Source data 1.** GFP quantification in ADL and ASJ neurons for *Figure 5A–D*.

**Source data 2.** Individual exploration assay results for *Figure 5E–G*.

**Figure supplement 1.** Tests of *srx-44* chemosensory activity.

**Figure supplement 1—source data 1.** ASH fluorescence (a.u.) values for buffer or icas#9 pulses.

icas#9, with stronger defects than eliminating either insulin (*daf-16*) or TGF-beta (*daf-3*) signaling (*Figure 5G*). These genetic results suggest that icas#9 foraging behavior, like dauer development, may involve convergence of insulin and TGF-β signaling onto the nuclear hormone receptor DAF-12.

Full-length SRX-44 translational fusions were localized to the sensory cilia of ADL (N2 and MY14 transgenes) and ASJ (MY14 transgenes) (*Figure 5—figure supplement 1A*), consistent with a sensory function for SRX-44. However, we could not detect acute icas#9 responses in ADL neurons using genetically-encoded calcium indicators (n = 7). This result is consistent with previous studies of the SRBC-64, SRX-43, SRG-36, and SRG-37 chemoreceptors, which detect ascaroside pheromones but do not elicit calcium transients in the ASK or ASI neurons in which they are normally expressed (*Kim et al., 2009*; *McGrath et al., 2011*). Ectopic expression of SRX-43, SRG-36, and SRG-37 in the ASH sensory neurons is sufficient to drive calcium transients in response to pheromones, but we were unable to elicit a response upon SRX-44 expression in ASH (*Figure 5—figure supplement 1B*), leaving its exact sensory properties unclear.

## Chemoreceptor genes are associated with balancing selection and positive selection in the *C. elegans* genome

Several studies have linked *C. elegans* chemoreceptor genes to natural or artificial selection. *srg-36* and *srg-37*, which encode chemoreceptors for the ascaroside ascr#5, have repeatedly mutated to inactivity in laboratory strains maintained under high density growth conditions (*McGrath et al., 2011*). *srbc-64*, which encodes a chemoreceptor for the ascaroside ascr#2, falls within a mating incompatibility locus that is under balancing selection (*Kim et al., 2009*; *Seidel et al., 2008*). These repeated associations raise the possibility that chemoreceptors are preferred substrates of *C. elegans* adaptation and evolution. In agreement with this hypothesis, chemoreceptor genes are enriched in regions that are hypervariable between the well-characterized N2 and CB4856 wild-type strains (*Thompson et al., 2015*). To ask if this observation extends across populations at a genomic level, we calculated the Tajima's D score for 20,037 5 Kb intervals spanning the entire genome using the sequences of 152 genetically distinct wild strains assembled by the Andersen lab at Northwestern University (*Cook et al., 2016*). Tajima's D is a statistical test used to determine if the DNA sequence is evolving by non-random processes. Regions of the genome that have recently undergone a selective sweep will have low values of Tajima's D, whereas regions of the genome under balancing selection will have high values of Tajima's D. Since population demographics such as recent population expansions or bottlenecks can also have a genome-wide effect on Tajima's D, we followed the approaches used in other species and focused our analysis on the bins with the most extreme high or low values. We found that across the genome, bins with low or high Tajima's D

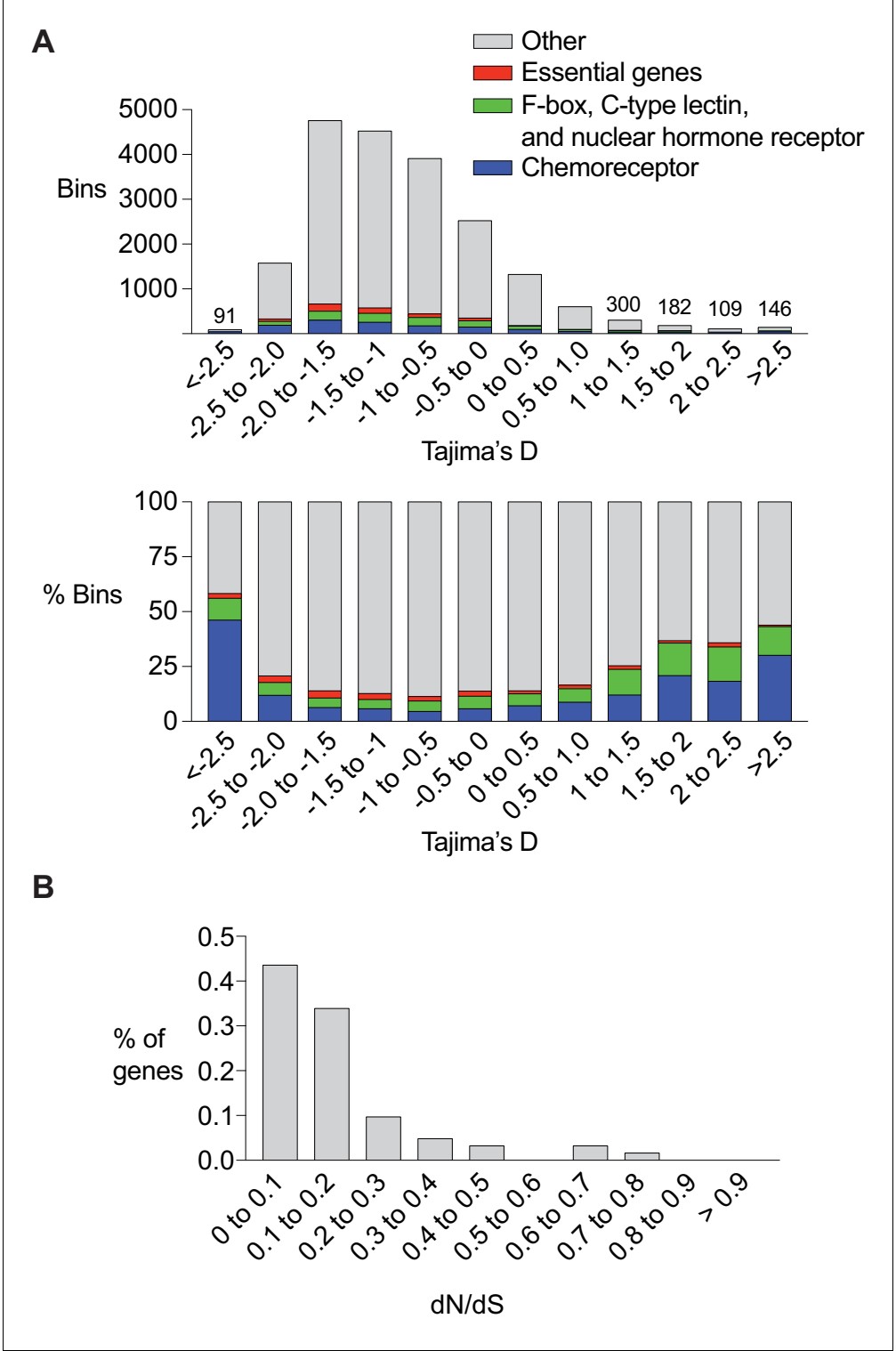

**Figure 6.** Chemoreceptor genes are targets of positive selection and balancing selection in *C. elegans*. (**A**) Enrichment of chemoreceptors among regions of the genome with low or high Tajima's D score. Top, total number of bins containing genes of different classes. For the highest and lowest Tajima's D (<5% of total), the numbers above bars indicate the total number of bins at that value. Bottom, percent of bins with a chemoreceptor. The *srx-43/srx-44* region within the *roam-1* locus has a Tajima's D of ~ 1. Bars are color coded, red = essential genes (*let, unc, dpy, bli, rol* and *egl* gene families), green = F-box, C-type lectin, and nuclear hormone

*Figure 6 continued on next page*

*Figure 6 continued*

receptors, blue = chemoreceptors, grey = other gene classes and bins with no genes. (**B**) dN/dS analysis of chemoreceptor genes in bins with a Tajima's D score above 1 (n = 62).

The following source data is available for figure 6:

**Source data 1.** Analysis of 5 kb bins spanning the genome for *Figure 6A*.
**Source data 2.** dN/dS values of chemoreceptor genes for *Figure 6B*.

score were enriched in chemoreceptor genes (*Figure 6A*). A remarkable fraction of the bins with the lowest and highest Tajima's D score contained chemoreceptor genes: 46.1% of bins with a score below −2.5 and 30.1% of the bins with a score above 2.5, representing a 6.59 and 4.30 fold enrichment respectively.

The chemoreceptor gene families of *C. elegans* are particularly large and diverse, with recent species-specific expansions even within the *Caenorhabditis* genus. As controls for these fast-evolving genes, we examined three other recently expanded *Caenorhabditis* gene families that encode F-box proteins, C-type lectins, and nuclear hormone receptors (*Thompson et al., 2015*). These genes were also enriched in bins with the lowest and highest Tajima's D scores, although less so than chemoreceptor genes (1.9 and 2.5 fold enrichment; *Figure 6A*). As a negative control, an analysis was conducted of genes likely to be under strong purifying selection, based on their phenotypes when mutated, the *let (lethal), unc (uncoordinated), dpy (dumpy), bli (blistered), rol (roller),* and *egl (egg-laying defective)* genes. These genes were depleted in the bins with the lowest and highest Tajima's D score, in agreement with their predicted properties (0.88 and 0.27 fold enrichment; *Figure 6A*). In summary, these results are consistent with the possibility that a substantial number of chemoreceptors – 100 genes or more – might be targets of balancing selection or selective sweeps, and that this may represent a substantial fraction of such selection detectable in the *C. elegans* genome.

A fraction of *C. elegans* chemoreceptor genes have idiosyncratic protein-disrupting mutations in subsets of wild-type strains, a result consistent with relaxed selection (*Stewart et al., 2005*; *Thomas and Robertson, 2008*). If the chemoreceptor divergence detected by Tajima's D reflects relaxed selection in one of these lineages, these genes should have a large number of protein-coding changes. dN/dS is a statistic that measures the evolutionary pressure on a protein using substitutions fixed along independent lineages. Because segregating polymorphisms (i.e. those used to calculate Tajima's D) do not represent fixed substitutions, care must be made in interpreting dN/dS ratios calculated from individuals within a population as dN/dS values can scale in unexpected ways (*Kryazhimskiy and Plotkin, 2008*). However, when balancing selection acts on two alleles for long evolutionary timescales, closely linked neutral sites will share the ancient coalescence time of the selected sites, creating two independent lineages upon which substitutions occur (*Charlesworth, 2006*). dN/dS can be applied in these situations, focusing on the substitutions that distinguish the two lineages. When we calculated dN/dS, we found that most of the chemoreceptor genes from bins with a high Tajima's D had a low dN/dS ratio. Among 62 genes with the strongest signature of balancing selection, defined by the existence of distinct haplotypes that included >50% of all known SNPs, 77% had a within-species dN/dS value less than 0.2 (*Figure 6B*), suggesting that both alleles remain under purifying selection. The low number of nonsynonymous mutations raises the possibility that noncoding variation in these genes drives functional diversity, as is the case with *srx-43* and *srx-44.* Together, these results implicate chemoreceptor genes as potential targets of selective pressures acting on noncoding sequence and gene expression.

## Discussion

The *roam-1* QTL for pheromone regulation of foraging behavior reflects changes in at least two genes. Previous experiments showed that reduced expression of the icas#9 receptor encoded by *srx-43* contributes to decreased pheromone sensitivity in wild strains with a MY14 *roam-1* haplotype (*Greene et al., 2016*). We find here that changes in *srx-44,* which encodes a closely-related chemoreceptor, also contribute to the *roam-1* QTL. The relationship between *srx-44* and behavior is not

simple: depending on its site of expression in either ADL or ASJ, *srx-44* enhances or inhibits the response to pheromones. Naturally occurring icas#9 insensitivity stems in part from increased expression of *srx-44* in ASJ, and in part from reduced *srx-43* expression in ASI.

It is intriguing that both *srx-43* and *srx-44* genes are present within a single QTL under balancing selection, suggesting that the combination of interacting mutations is biologically significant. Co-selection of multiple genes is a substrate for novel functions in evolution, and close linkage can maintain these combinations as they spread through a population. The importance of linked 'supergenes' in balancing selection has long been recognized in flowering plants, where tightly linked genes that control flower morphology give rise to distinct mating groups within a species (*Barrett and Shore, 2008*; *Ernst, 1955*). Other polymorphic supergenes are responsible for color mimicry in *Heliconius* butterflies and single- versus multiple-queen colony organization in fire ants (*Joron et al., 2011*; *Wang et al., 2013*). With its low recombination rate, the predominantly self-fertilizing *C. elegans* is likely permissive for the accumulation of haplotypes with functionally and genetically linked mutations, including *srx-43/srx-44* and the two-gene mating incompatibility locus *zeel-1/peel-1* (*Seidel et al., 2008*). The extraordinarily high sequence divergence between *roam-1* alleles may further stabilize distinct N2 and MY14 haplotypes by suppressing local meiotic recombination. Similar examples of functional linkage may be represented in other hyper-divergent haplotypes in the *Caenorhabditis* genome, of which there are 61 between the commonly studied N2 and CB4856 strains (*Thompson et al., 2015*).

The activity of *srx-44* is only detectable in the presence of a functional *srx-43* gene and icas#9. *srx-43* encodes a biochemically active icas#9 receptor, but we were unable to demonstrate a similar activity for *srx-44*. The sequence similarity between SRX-43 and SRX-44 suggests that both could recognize pheromones, but it is possible that SRX-44 has a modulatory effect that is not associated with direct pheromone binding. Alternatively, SRX-44 may detect icas#9, but require a coreceptor that is not present in the neuron used for ectopic testing (ASH), or a signaling pathway that is poorly coupled to calcium signaling. Indeed, many pheromone responses in *C. elegans* are mediated by changes in gene expression rather than acute calcium signaling (*Kim et al., 2009*; *Li et al., 2003*; *Vowels and Thomas, 1992*), and the same appears to be true of the icas#9 effect on foraging: the pheromone regulates both insulin and TGF-beta peptide genes, and requires at least six hours to alter foraging, consistent with a slow transcriptional regulatory pathway (*Greene et al., 2016*). This slow action is characteristic of many animal pheromones that alter physiology or reproduction (priming pheromones), rather than eliciting immediate behaviors (*Conte and Hefetz, 2008*; *Sorensen et al., 1989*).

The effect of *srx-44* loss of function depends on the genotype of the *roam-1* region: inactivating the gene in the *roam-1*$_{N2}$ background decreased the pheromone response, but inactivating the gene in the *roam-1*$_{MY14}$ background increased the pheromone response. This is an example of sign epistasis, when the effect of one mutation (the *srx-44* null) has the opposite effect in the presence of another mutation (the *roam-1* QTL)(*Phillips, 2008*; *Weinreich et al., 2005*). Due to the ubiquity of epistasis in natural traits (*Buchner and Nadeau, 2015*), there is general interest in understanding the mechanisms that cause non-linear interactions between genetic variation in natural populations. The epistasis we observe here can be explained as a consequence of genetic variation that changes the site of *srx-44* expression, which in turn modifies the role of *srx-44* on the trait in question. The opposing effects of the ASJ and ADL neurons explain the genetic epistasis between *srx-44* null mutations and the natural variants that affect *srx-44* expression. We refer to this mechanism of epistasis as 'cellular epistasis.' Superimposed on this effect of *srx-44* is antagonism between *srx-43* in ASI and *srx-44* in ASJ, which creates another level of cellular epistasis in the *roam-1*$_{MY14}$ background. Given the importance of *cis*-regulatory changes in natural traits, cellular epistasis could play an underappreciated role in the epistasis observed in natural traits.

Sensory receptors evolve rapidly between species based on their ecological needs (*Bargmann, 2006*; *Nei et al., 2008*). These families frequently gain and lose members by duplication and divergence, and can acquire new properties at the level of protein function. For example, the G protein-coupled receptors for sweet taste have been lost several times independently in carnivorous vertebrates such as cats, reptiles, and bats, which lack sugar in their diet (*Jiang et al., 2012*; *Zhao et al., 2010*); the reappearance of a sugar-rich diet in nectar-feeding birds was accompanied by the elaboration of new sweet taste receptors through functional divergence of an amino acid taste receptor (*Baldwin et al., 2014*). Unlike these examples of altered chemoreceptor proteins, the

*roam-1* QTL is associated with changes in gene expression without changes in gene number or apparent protein activity. *srx-43* and *srx-44* are expressed and have biological activity in both N2 and MY14 strains; the *roam-1* QTL respecifies behaviors by modifying the levels and sites of receptor expression. Gene expression changes are common targets of evolutionary adaptation in development (*Chan et al., 2010*; *Sucena et al., 2003*); this work suggests that a similar principle may apply to chemoreceptor genes.

Studies of chemosensation in *C. elegans* revealed the importance of a labeled-line system, in which individual sensory neurons are strongly linked to particular behavioral outputs such as attraction and avoidance (*Troemel et al., 1997*). Subsequent studies have demonstrated that labeled lines are also present in the mammalian taste system (*Mueller et al., 2005*), somatosensory system (*Han et al., 2013*), and parts of the olfactory system (*Kobayakawa et al., 2007*). In a labeled line system, an existing receptor for a biologically relevant molecule can be respecified to a new behavior by changing its pattern of expression to a different sensory neuron. In an individual's lifetime, such plasticity can drive rapid changes in behavioral responses to a stimulus (*Ryan et al., 2014*). Our results show that changes in chemoreceptor expression are also significant on evolutionary timescales, suggesting that chemoreceptor expression changes could be a potent and effective mechanism by which natural variation reshapes innate behaviors and physiology.

## Materials and methods

### Nematode culture

Strains were grown under standard conditions at 21–22°C on Nematode Growth Medium (NGM) plates seeded with *Escherichia coli* OP50 bacteria.

### Strains

#### Parental strains

Wild-type *roam-1*$_{N2}$ and *roam-1*$_{MY14}$ strains were:

N2, Bristol isolate

CX16300 *kyIR163 [roam-1*$_{MY14}$ V: ~ 15.861- ~ 16.043 Mb, MY14>N2]* (*Greene et al., 2016*).

This strain has 182 kb derived from the MY14 strain introgressed into an N2 genetic background.

#### Mutants

New mutations were backcrossed into the N2 or *kyIR163* genetic background 4-7 times before characterization.

CX16849 *srx-43(gk922634) V*. R160opal, generously provided by the Million Mutation project.

CX16935 *kyIR163 srx-43(ky1019) V*. ky1019 is a CRISPR-generated insertion/deletion leading to a frameshift and early termination (*Greene et al., 2016*).

CX16490 *srx-44(ky1009) V*. ky1009 is a CRISPR-generated two bp deletion (TACCCTTC–GTTCTTGATA) leading to a frameshift after amino acid 57.

CX16934 *kyIR163 srx-44(ky1013) V*. ky1013 is a CRISPR-generated two bp deletion (TATGCGCCGG–ACAACGACAGATT) leading to a frameshift after amino acid 49.

CX16936 *kyIR163 srx-43(ky1013) srx-44(ky1023) V*. ky1023 is a CRISPR-generated frameshift after amino acid 98 (3 bp deletion (CCG), 7 bp insertion (ggaattc), final sequence (CTCTCAAGTCCggaattcTTCGGA)).

CB1370 *daf-2(e1370ts) III*

CX13568 *daf-28(sa191) V*

CX17338 *daf-28(tm2308) V*

JT5464 *daf-7(e1372) III; daf-3(e1376) X*

DR26 *daf-16(m26) I*

DR20 *daf-12(m20) X*

#### MosSCI strains

*kySi66* is a Mos-1 Single Copy Insertion (MosSCI) of the N2 *srx-43* genomic sequence in the landing site *ttTi5605 II. unc-119* animals containing the *ttTi5605* landing site were injected with a mix of plasmids containing transposase, *srx-43* and *Cbr-unc-119* within flanking homology arms, and negative

selection markers (*myo-3::mcherry, myo-2::mcherry, hsp::peel-1*). Progeny were heat shocked, and after 24 hr, coordinated animals without *mcherry* signal were allowed to lay eggs, and subsequent generations were genotyped to confirm single copy insertion of the transgene (*Frøkjær-Jensen, 2015*).

 CX17196 *kySi66 II; srx-43(gk922634) V*
 CX17260 *kySi66 II; kyIR163 srx-43(ky1013) V*

## Transgenic lines

Transgenic lines were generated by injecting the syncytial gonal of adult hermaphrodites with the desired transgene, a fluorescent co-injection marker, and empty vector bringing the total DNA concentration to 100 ng/ul. To control for the variability of extrachromosomal transgenic line, three independent lines were tested for each transgene.

 CX16431 *kyIR163 V; kyEx5600 [Psrx-44(N2)::srx-44(N2)::sl2::GFP 2.5 ng/ul, Pmyo-3::mcherry 5 ng/ul]*

 CX16480 *kyIR163 V; kyEx5626 [Psrx-44(MY14)::srx-44(MY14)::sl2::GFP 2.5 ng/ul, Pmyo-3::mcherry 5 ng/ul]*

 CX17339 *kyIR163 V; kyEx6058 [srx-44(lf) 2.5 ng/ul, Pmyo-3::mcherry 5 ng/ul]*

 CX16477 *kyIR163 V; kyEx5623 [Psrx-44(MY14)::srx-44(N2)::sl2::GFP 2.5 ng/ ul, Pmyo-3::mcherry 5 ng/ul]*

 CX16483 *kyIR163 V; kyEx5629 [Psrx-44(N2)::srx-44(MY14)::sl2::GFP 2.5 ng/ ul, Pmyo-3::mcherry 5 ng/ul]*

 CX16624 *kyIR163 V; kyEx5698 [Psrx-44(N2)::GFP 2.5 ng/ul, Pmyo-3::mcherry 5 ng/ul]*

 CX16627 *kyIR163 V; kyEx5701 [Psrx-44(MY14)::GFP 2.5 ng/ul, Pmyo-3::mcherry 5 ng/ul]*

 CX16752 *kyIR163 V; kyEx5775 [Psrx-44(N2 proximal, MY14 distal)::GFP 2.5 ng/ul, Pmyo-3:: mcherry 5 ng/ul]*

 CX16735 *kyIR163 V; kyEx5759 [Psrx-44(N2 proximal, MY14 distal)::GFP 2.5 ng/ul, Pmyo-3:: mcherry 5 ng/ul]*

 CX16425 *kyIs602 [Psra-6:GCaMP3.0 75 ng/uL, coel:GFP 10 ng/uL, integrated]; kyEx5594 [Psra-6:: srx-44(N2) 50 ng/ul, Pmyo-3::mcherry 5 ng/ul]*

 CX16428 *kyIs602; kyEx5597 [Psra-6::srx-44(MY14) 50 ng/ul, Pmyo-3::mcherry 5 ng/ul]*

 CX16533 *kyIR163 V; kyEx5646 [Psrh-11(ASJ)::srx-44::sl2::GFP 50 ng/ul, Pmyo-3::mcherry 5 ng/ul]*

 CX16530 *kyIR163 V; kyEx5643 [Psrh-220(ADL)::srx-44::sl2::GFP 50 ng/ul, Pmyo-3::cherry 5 ng/ul]*

 CX16593 *kyEx5670 [Psre-1(ADL)::TeTx 50 ng/ul, myo-3::cherry 5 ng/ul]*

 CX16611 *kyIR163 V; kyEx5688 [Psrh-11(ASJ)::TeTx 50 ng/ul, myo-3::cherry 5 ng/ul]*

 CX14365 *kyEx4565 [ Psrg-47(ASI)::TeTx::sl2mcherry 50 ng/ul, myo-3::GFP 25 ng/ul]*

 CX17341 *srx-44(ky1009) V; kyEx5670*

 CX17340 *srx-44(ky1009) V; kyEx5688*

 GR1455 *mgIs40 [Pdaf-28::GFP], outcrossed 5X*

 CX16956 *kyIR163 V; mgIs40*

 CX16957 *srx-43(gk922634) V; mgIs40*

 CX15046 *lin-15(n765) X; kyIs128 [lin-15, Pstr-3::GFP]*

## CRISPR allele replacement strains:

Allele replacement lines for the *srx-44* promoter were created using an established coCRISPR protocol (*Arribere et al., 2014*). Young adults with just a few fertilized eggs were injected in the syncytial gonad with (i) a mix of plasmids encoding Cas9, guideRNA for the target site, and guideRNA for *rol-6*, and (ii) ssDNA homologous recombination templates for inducing a dominant *rol-6* mutation and the desired promoter substitution. F1 roller progeny were screened for the promoter allele replacement by sequencing after laying eggs, and then backcrossed four times into the parental strain.

 CX16800 *ky982 V. ky982* replaces 38 bp in the N2 region upstream of *srx-44* with 32 bp from MY14(TATCATTTAAAAACTCACTTTTTGAGTAGTCG replaces AATCATGTTAAAAAATCAA TTTTTGGGTAGTCAACGGA).

 CX16799 *kyIR163 V; ky1015 V. ky1015* replaces 32 bp in the MY14 region upstream of *srx-44* with 38 bp from N2 (AATCATGTTAAAAAATCAATTTTTGGGTAGTCAACGGA replaces TATCA TTTAAAAACTCACTTTTTGAGTAGTCG).

## Behavioral analysis

Exploration assays were used to determine the effect of pheromone on foraging behavior (*Flavell et al., 2013*). A single L4 hermaphrodite was placed on an evenly seeded 3.5 cm plate containing 10 nM icas#9, a concentration that can accumulate in a high-density population of *C. elegans* (*Greene et al., 2016*), or control solvent. After a 16 hr period, the assay was quantified by placing the plate on a grid with 86 squares and counting the number of squares entered by the animal (*Figure 1A*). Pheromone response was then calculated as the difference in squares entered on pheromone plates and control plates (*Figure 1B*). A positive pheromone response indicated that animals entered fewer squares in the presence of icas#9.

## Imaging

All GFP imaging was conducted on adult animals mounted on 2% agarose pads and immobilized with 5 mM sodium azide. Images were collected with a 100X objective on a Zeiss Axio Imager.Z1 Apotome microscope (Carl Zeiss, Oberkochen, Germany) with a Zeiss AxioCam MRm CCD camera. For GFP expression studies, GFP levels were quantified 16 hr after L4 animals were placed on exploration assay plates with or without added icas#9. Images were processed in Metamorph (Molecular Devices, Sunnyvale, CA). ImageJ was used to create a maximum intensity Z-projection and assess the reporter value, which was determined as the mean gray value for a 16-pixel-radius circle centered over the cell body minus the mean background intensity.

Calcium imaging experiments were performed and analyzed as described previously (*Larsch et al., 2013* ). Young adult animals were placed into custom-made 3×3 mm microfluidic polydimethylsiloxane devices, permitting rapid changes in stimulus solution, and containing two arenas allowing for simultaneous imaging of two genotypes with approximately ten animals each. Animals were transferred to the arenas in S-Basal buffer and paralyzed for 80–100 min in 1 mM (-)-tetramisole hydrochloride. Experiments consisted of four 10 s pulses of stimulus separated by 30 s of buffer, with 60 additional seconds between stimulus types. Tiff stacks were acquired at 10 frames/second at 5x magnification on a Hamamatsu Orca Flash 4 sCMOS (Hamamatsu Photonics, Sunayama-cho, Japan), with 10 ms pulsed illumination every 100 ms with a Sola set at 470/40 nm excitation (Lumencor, Beaverton, Orgeon, USA).

## Genomic analysis

The current version of the VCF file (WI.20160408.snpeff.vcf.gz) containing SNVs for all wild *C. elegans* isolates was downloaded from the CENDR website (www.elegansvariation.org) (*Cook et al., 2016*). From this file, we calculated Tajima's D on 5 kb bins using vcftools (*Danecek et al., 2011*). To determine if this bin contained a chemoreceptor gene, we used the gff3 file provided by www.wormbase.org for the WS245 release, searching for gene names that start with known chemoreceptor families as determined by the following reference (*Thomas and Robertson, 2008*). To determine if the bin contained a fast-evolving gene family, we searched for gene names starting with *fbxa, fbxb, fbxc, clec,* or *nhr*. To determine if the bin contained a gene likely to be under purifying selection, we searched for gene names that started with *unc, bli, let, dpy, rol,* or *egl*.

To calculate dN/dS, we identified 62 chemoreceptors with the most pronounced signatures of balancing selection in two independent lineages. This signature was defined as chemoreceptors with >50% of the polymorphisms detectable in CeNDR (*Cook et al., 2016*) present on a shared haplotype. Using these polymorphisms, we calculated dN/dS by a counting method using transcript sequence files downloaded from the WS245 version of Wormbase along with annotations as determined by SnpEff (*Cingolani et al., 2012*) included in the downloaded VCF file. For 77% of these chemoreceptor genes, dN/dS was <0.2, suggesting that they were recently under purifying selection; previous studies suggest that chemoreceptor dN/dS ratios between species and between closely related paralogs are ~0.2 (*Stewart et al., 2005*).

## Statistics

Most experiments were repeated on three or four separate days. For exploration assays, the standard group size was six per day and all plates with a healthy adult animal at the end of the assay were scored and included in the analysis. To ensure randomization, the following or a similar approach was used: at the start of each exploration assay, six animals were placed on a pick at a

time, and in the order the animals came off the pick, they were then transferred individually to three control plates and then to three icas#9 plates. Scoring of assays was conducted by an experimenter blind to the condition or genotype.

Most statistical comparisons were done by ANOVA with Dunnett correction for multiple comparisons or (two-sided) t test, as noted in the figure legends. The normality of the data was tested usually with D'Agostino-Pearson omnibus test. The Bartlett's test was used to check for differences in variance between groups being statistically compared.

## Acknowledgements

We thank Erik Andersen for sharing the unpublished CeNDR database, the *Caenorhabditis* Genetics Center (CGC), which is supported by NIH grant P40 OD010440, and the Million Mutation project for strains, and Elias Scheer for comments on the manuscript. RAB was supported by the Research Corporation for Science Advancement (Cottrell Scholar Award, 22844) and the NIH (GM118775). PTM was supported by NIH grant R01GM114170 and the Ellison Medical Foundation. JG was supported by the NIH grant F30 MH101931-03. CIB was an investigator of the HHMI. This work was supported by the Ellison Medical Foundation.

## Additional information

### Funding

| Funder | Grant reference number | Author |
|---|---|---|
| National Institutes of Health | Graduate student fellowship | Joshua S Greene |
| National Institutes of Health | F30 MH101931-03 | Joshua S Greene |
| National Institutes of Health | GM118775 | Rebecca A Butcher |
| Research Corporation for Science Advancement | Cottrell Scholar Award, 22844 | Rebecca A Butcher |
| Ellison Medical Foundation | Investigator | Patrick T McGrath Cornelia I Bargmann |
| National Institutes of Health | R01GM114170 | Patrick T McGrath |
| Howard Hughes Medical Institute | Investigator | Cornelia I Bargmann |

The funders had no role in study design, data collection and interpretation, or the decision to submit the work for publication.

### Author contributions

JSG, Conception and design, Acquisition of data, Analysis and interpretation of data, Drafting or revising the article; MD, PTM, Acquisition of data, Analysis and interpretation of data, Drafting or revising the article; RAB, Synthesized pheromones, Contributed unpublished essential data or reagents; CIB, Conception and design, Analysis and interpretation of data, Drafting or revising the article

### Author ORCIDs

Rebecca A Butcher, http://orcid.org/0000-0002-0925-4459
Cornelia I Bargmann, http://orcid.org/0000-0002-8484-0618

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
