## [Decision Letter]

Thank you for submitting your article "Expression remapping of two chemoreceptor genes in a *Caenorhabditis elegans* QTL for foraging behavior" for consideration by *eLife*. Your article has been reviewed by two peer reviewers, including David L Stern (Reviewer #1), and the evaluation has been overseen by a Reviewing Editor and a Senior Editor.

The reviewers have discussed the reviews with one another and the Reviewing Editor has drafted this decision to help you prepare a revised submission.

Summary:

This is a truly excellent contribution. The authors have determined that a previously-identified QTL for pheromone response in *C. elegans*, where the authors previously identified one locus under balancing selection, actually contains a second closely linked and functionally similar gene that also harbors variation for this behavior. The work is comprehensive and wide-ranging, including detailed molecular analysis that identifies the key variable region as causing a change in expression pattern, exploration of the effect of the pheromone on transcription of key genes and signaling pathways, and a genome-wide evolutionary analysis that suggests that selection may be acting differentially on genes encoding pheromone sensors.

The identification of natural polymorphisms that influence behavior has been challenging, with only a very limited number of cases. The work of Greene et al. goes beyond simply the challenge of identifying causative polymorphisms to further corroborating an increasing number of cases that point to chemoreceptor expression as an important source of genetic variation on which natural selection can act. Thus, the work will be of broad interest to investigators interested in the intersection of evolutionary biology and behavior.

Essential revisions:

1) Please expand the discussion and analysis of dN/dS. The authors utilize the analysis in the context of sequences compared from populations within a species, which are not necessarily fixed independently, rather than in the context of the analysis of genomes from divergent distinct species. Such circumstances make the discussion and application of dN/dS measurements more challenging to interpret. Further discussion of how these caveats may influence the analysis of sequences among different strains of *C. elegans* may be helpful.

2) Figure 5 and subsection “icas#9 influences behavior through TGF-β and insulin signaling pathways”, second paragraph: Please clarify the text and the model. The authors should walk the readers through each piece of evidence that illustrates autonomy versus non-autonomy. It would be helpful to break up 5A into four separate panels, so that they can be referred to specifically when explaining the results. Similarly, more explanation of the data that supports the arrows implying activation and repression (of genes? of neural activity?) in Figure 5 is required. The TeTx labels in the diagram are confusing and should be removed.

3) The term "expression remapping" in the title is confusing. A more descriptive title for a general audience would be useful.

---

## [Author Response]

*[…] Essential revisions:*

*1) Please expand the discussion and analysis of dN/dS. The authors utilize the analysis in the context of sequences compared from populations within a species, which are not necessarily fixed independently, rather than in the context of the analysis of genomes from divergent distinct species. Such circumstances make the discussion and application of dN/dS measurements more challenging to interpret. Further discussion of how these caveats may influence the analysis of sequences among different strains of C. elegans may be helpful.*

We have expanded our discussion of dN/dS (subsection “Chemoreceptor genes are associated with balancing selection and positive selection in the *C. elegans* genome”, last paragraph), and include the following statement: “dN/dS is a statistic that measures evolutionary pressure on a protein using substitutions fixed along independent lineages. […] Among 62 genes with the strongest signature of balancing selection, defined by the existence of distinct haplotypes that included >50% of all known SNPs, 77% had a within-species dN/dS value less than 0.2 (Figure 6), suggesting that both alleles remain under purifying selection.”

*2) Figure 5 and subsection “icas#9 influences behavior through TGF-β and insulin signaling pathways”, second paragraph: Please clarify the text and the model. The authors should walk the readers through each piece of evidence that illustrates autonomy versus non-autonomy. It would be helpful to break up 5A into four separate panels, so that they can be referred to specifically when explaining the results. Similarly, more explanation of the data that supports the arrows implying activation and repression (of genes? of neural activity?) in Figure 5 is required. The TeTx labels in the diagram are confusing and should be removed.*

We have divided the initial Figure 5 into separate panels, 5A-D, and expanded our discussion of the evidence for non-cell-autonomous action of each receptor in the second paragraph of the subsection “icas#9 influences behavior through TGF-β and insulin signaling pathways”.

We have also expanded our description of the evidence for the arrows in 5H (initially Figure 5) in the figure legend, and removed tetanus toxin from the figure.

*3) The term "expression remapping" in the title is confusing. A more descriptive title for a general audience would be useful.*

We have changed the title to “Regulatory changes in two chemoreceptor genes contribute to a *Caenorhabditis elegans* QTL for foraging behavior”